# Evaluation of fetal exposure to environmental noise using a computer-generated model

Pierre Gélat [1], Elwin van't Wout [2], Reza Haqshenas[3], Andrew Melbourne[4,5], Anna L. David [5], Nada Mufti[5], Julian Henriques[6], Aude Thibaut de Maisières[7] & Eric Jauniaux[5]

Acoustic noise can have profound effects on wellbeing, impacting the health of pregnant women and their fetus. Mounting evidence suggests neural memory traces are formed by auditory learning in utero. A better understanding of the fetal auditory environment is therefore critical to avoid exposure to damaging noise levels. Using anatomical data from MRI scans of pregnant patients ($N = 4$) from 24 weeks of gestation, we develop a computational model to quantify fetal exposure to acoustic field. We obtain acoustic transfer characteristics across the human audio range and pressure maps in transverse planes passing through the uterus at 5 kHz, 10 kHz and 20 kHz, showcasing multiple scattering and modal patterns. Our calculations show that the sound transmitted in utero is attenuated by as little as 6 dB below 1 kHz, confirming results from animal studies that the maternal abdomen and pelvis do not shelter the fetus from external noise.

We are the first humans to globally expose with our activities the next generations to major climate changes and environmental pollution before they are born[1]. The main types of pollution are usually classified by environment and include air pollution, water pollution and land pollution. Noise pollution resulting from environmental noise (road traffic, railway and aircraft noise, wind turbine noise, occupational and leisure noise) has been identified as a growing concern for the long-term impacts on physical and mental human health by both the World Health Organisation[2] and the European Union[3]. Over the last two decades, observational and experimental studies have shown that noise exposure increases the occurrence of hypertension and cardiovascular disease, disturbs sleep and causes daytime sleepiness and affects patient outcomes and staff performance in hospitals[4]. Epidemiological data have also shown that noise exposure in early life impairs cognitive performance and motor function in children and preadolescents[5,6]. The effects of occupational noise, which is often in the range of between 80 and 100 dB, on hearing loss[7] and hypertension[8] are now well established. A model for assessing traffic noise exposure in the London area estimated that the equivalent continuous traffic noise level over the period 07:00–23:00 h, was between 55 and 83 dB(A)[9]. The association between exposure to road traffic noise and ischemic heart disease has also been well established[10,11]. Whilst increased exposure to air pollutants and particulate emissions is a likely contributor to this association, a recent compendium has provided an overview of epidemiological research on the effects of transportation noise on cardiovascular disease and associated risk factors[12]. Based on the outcomes of experimental and clinical studies reviewed by Münzel et al.[12], mechanistic insights are provided, with the potential effects of noise on vascular dysfunction, oxidative stress, and inflammation in both humans and animals. A recent report from the European Environment Agency has shown that long-term exposure to environmental noise is estimated to cause

[1]Department of Surgical Biotechnology, Division of Surgery and Interventional Science, University College London, London, UK. [2]Institute for Mathematical and Computational Engineering, Pontificia Universidad Católica de Chile, Santiago, Chile. [3]Department of Mechanical Engineering, University College London, London, UK. [4]School of Biomedical Engineering & Imaging Sciences, Faculty of Life Sciences & Medicine, King's College London, London, UK. [5]Elizabeth Garrett Anderson Institute for Women's Health, University College London, London, UK. [6]Department of Media, Communications and Cultural Studies, Goldsmiths University of London, London, UK. [7]Sonic Womb Productions Limited, London, UK. ✉e-mail: p.gelat@ucl.ac.uk

12,000 premature deaths and contribute to 48,000 new cases of ischemic heart disease per year in the European territory[13].

Over the last two decades, there has been mounting epidemiological and basic science evidence showing the impact of climate change[14] and air pollution[15] on pregnancy outcomes. Ambient black carbon particles and microplastics have been identified in the intracellular compartment of human placentas[16,17] and recently in fetal organs[18], suggesting a direct fetal exposure to these pollutants before birth. However, data on the effects of environmental noise on pregnancy, birth and reproductive outcomes are limited[19–22]. Road traffic noise has been associated with maternal weight gain during and after the pregnancy[23], whereas railway noise may be associated with gestational diabetes mellitus[24]. Regarding the direct effect of environmental noise on fetal development, there is no evidence showing an increased risk of congenital malformations and the evidence for the association between road noise and fetal growth is uncertain with only some studies showing a moderate effect. A recent study has shown that occupational noise exposure during pregnancy to 80–85 dB(A) of annual average 8-h occupational noise level in 5-year intervals is also associated with an increased risk of all pregnancy-related hypertension whereas exposure to >85 dB(A) of noise is, as with railway noise, also associated with an increased risk of gestational diabetes mellitus[25]. A Swedish nationwide cohort study has shown an association between occupational noise during pregnancy and hearing dysfunction in children[26]. The association was strongest for mothers who worked full time during pregnancy and were exposed to >85 dB(A) equivalent continuous noise level over an 8-hour period. These data suggest a direct effect of occupational noise exposure on the human fetus. The main concern is during the third trimester of pregnancy when the fetal brain structural and functional changes occur rapidly and are shaped by sensory inputs and endogenous neural activity with a direct impact on speech processing before birth[27,28]. A review of the known effects of environment and occupational noise during pregnancy on perinatal and maternal outcome nevertheless concluded that further studies are required so that the effects of both occupation and environmental noise exposure on obstetric patients may be underpinned[22].

Unlike fetal exposure to air or water pollutants which can be directly evaluated by sampling tissues and body fluids, there are limited in vitro and in vivo models to study human fetal exposure to environmental noise. In vivo experiments in sheep and goats using hydrophone recordings have indicated that intra-uterine noise is predominantly low-frequency[29] and exposure to intense broadband noise altered the fetal auditory brain stem response and damaged cochlea hair cells[30]. These experimental data are limited by the quality of recording technology and access to computer models enabling the translation of animal data into information about humans. Using modern acquisition systems and calibrated instrumentation to measure the in-utero acoustic transfer characteristics on pregnant ewes, we found that frequency content above 10 kHz is transmitted into the amniotic sac, and that some frequencies are attenuated by as little as 3 dB[31]. However, translating experimental data obtained on ovine models into humans remains challenging due to fundamental anatomical differences between both species. Furthermore, the physiologies of the respective uterine environments differ. In vivo measurement of the sound field in humans presents ethical challenges. Another consideration for moving beyond in vivo experiments includes the fact that, using acoustic instrumentation, field quantities can only be monitored at a very limited number of physical locations.

To completely map an in-utero sound field in 3D would require multiple in vivo measurements beyond what is physically practicable. It is therefore desirable to seek solutions to this problem beyond in vivo measurement by attempting to predict the physical propagation of acoustic waves inside the pregnant woman. In this work, we aim to substantiate the extent of in-utero sound transmission by using a computational model for studying fetal exposure to external sound sources, including environmental, leisure, and occupational noise. Based on prior in vivo work on sheep[29], we hypothesize that the maternal abdomen and other anatomical groups do not acoustically isolate the developing fetus from the external sonic environment. Furthermore, we anticipate that as the excitation frequency increases across the human audio range and the wavelengths in tissue are of the order or less than the anatomical dimensions, a complex acoustic in-utero environment materializes where modal behavior and multiple scattering are observed.

We imported anatomical data obtained from MRI scans on selected pregnant women at specific stages of gestation to predict the in-utero acoustic field as a function of external acoustic excitation throughout the audio range (20 Hz–20 kHz). Using mathematical formulations based on the boundary element method (BEM)[32–34] as implemented in the open-source OptimUS Python library[35] we predict the sound pressure level (SPL) throughout the volume of the uterus for a unit amplitude plane wave normally incident on the maternal abdomen. The employed BEM formulation features the capability of producing accurate results in scenarios where interfaces between two media feature a large acoustic impedance contrast, defined as the product of the speed of sound with the density (such as an air/soft tissue interface)[36]. Furthermore, our numerical scheme makes it possible to carry out calculations in cases where the dimensions of the computational domain are large relative to the acoustic wavelengths involved, as is the case for in-utero sound transmission towards the higher end of the audio range. OptimUS was validated against ten different numerical modeling techniques for acoustic propagation prediction in the context of a transcranial ultrasound computational benchmarking exercise, including the finite-difference time-domain method, angular spectrum method, pseudospectral method, and spectral-element method[37].

## Results

### Anatomical data

We acquired MRI data from 4 singleton pregnancies. The datasets are referred to as Subject 1, Subject 2, Subject 3, and Subject 4. The gestational age for each dataset was as follows:

- Subject 1: 25 weeks and 1 day
- Subject 2: 32 weeks and 1 day
- Subject 3: 36 weeks and 2 days
- Subject 4: 37 weeks and 3 days.

Given that the wavelength in soft tissue at 20 kHz is approximately 75 mm, we assumed that the resolution of the MRI scans (0.74 × 0.74 mm) as well as the transformation of the raw data via smoothing algorithms will not generate significant uncertainties in the SPL predictions at the frequencies of interest. To produce a realistic model of in utero sound propagation for compressional waves, we focussed on the critical tissue paths resulting from the abdominal region, i.e., the uterus, and the spine. The uterine region comprises the uterine wall, the fetus and the presence of amniotic fluid. We considered the uterus to be composed of either amniotic fluid (low attenuation case) or muscle tissue (high attenuation case) rather than providing a model with detailed attenuation parameters. These scenarios served as a worst-case and best-case scenario, respectively, for in-utero acoustic transmission. We also considered the presence of the maternal spine. Whilst its diameter is small relative to the wavelengths of interest, it features significant acoustic contrast with soft tissue. We considered the regions of the abdominal section, not including the maternal spine or the uterus to be filled with generic soft tissue. The meshes of the anatomical domains corresponding to the abdomen, the uterus, and the spine are shown in Fig. 1A–D, for Subjects 1, 2, 3, and 4, respectively.

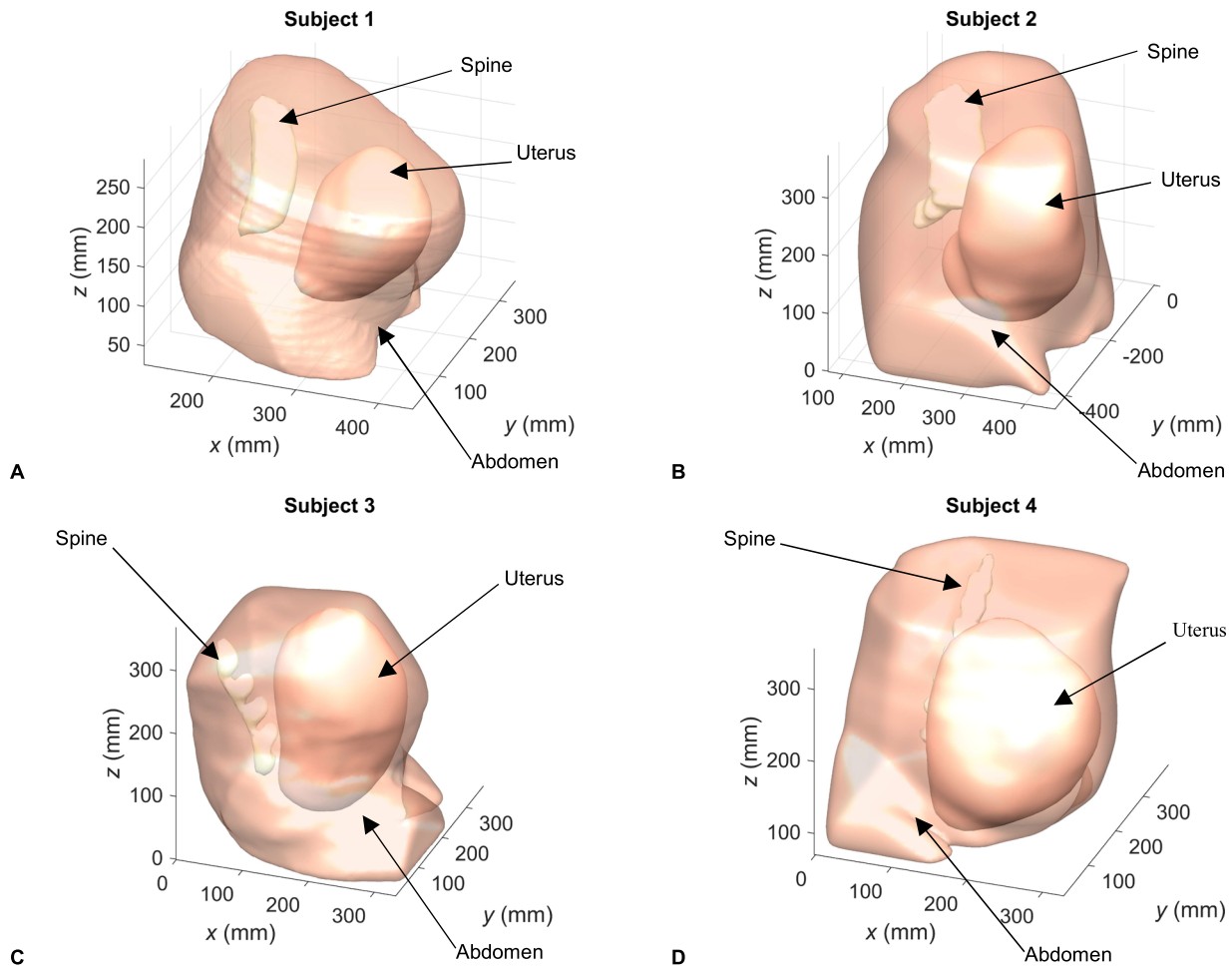

**Fig. 1 | Anatomical regions of datasets used in computational meshes.** This figure shows the surface boundaries of the three anatomical regions considered for datasets used in computational meshes: **A** Subject 1, **B** Subject 2, **C** Subject 3, and **D** Subject 4. The anatomical regions are the maternal abdomen, the spine and the uterus.

## Material properties

The soft tissue and bone regions are treated as piecewise homogeneous acoustic domains. Whilst the speed of sound of compressional waves in soft tissue has been characterized at audio range frequencies, there is limited information on the attenuation coefficient of compressional waves at these frequencies. Thus, the attenuation coefficient for the tissue groups of interest was estimated from viscoelastic measurements on ex vivo tissue. In an infinite viscoelastic material, the speed of sound of longitudinal waves may be expressed as:

$$c_L = \sqrt{\frac{E(1 + i\tan\delta)}{\rho}}, \tag{1}$$

where $E$ is Young's modulus, $\tan\delta$ is the loss tangent, $\rho$ is the density and $i$ is the imaginary unit. In the absence of shear waves, we approximate Young's modulus to the bulk modulus $K$, which is given by:

$$K = \rho c_0^2, \tag{2}$$

where $c_0$ is the equilibrium speed of sound in the medium. Equation (1) then becomes:

$$c_L = c_0\sqrt{1 + i\tan\delta} = c_0\sqrt{\frac{\cos\delta + i\sin\delta}{\cos\delta}} = \frac{c_0}{\sqrt{\cos\delta}}e^{\frac{i\delta}{2}}, \tag{3}$$

The complex wave number is:

$$k = \frac{\omega}{c_L} = \frac{\omega\sqrt{\cos\delta}}{c_0}\left(\cos\frac{\delta}{2} - i\sin\frac{\delta}{2}\right), \tag{4}$$

Hence, the attenuation coefficient $\alpha$ is given by:

$$\alpha(f) = \frac{2\pi\sqrt{\cos\delta}}{c_0}\sin\frac{\delta}{2}f, \tag{5}$$

It should be noted that since $\delta$ is small, the wavenumber may be approximated by:

$$k = \frac{\omega}{c_L} = \frac{\omega}{c_0}\left(1 - i\frac{\delta}{2}\right), \tag{6}$$

Values for $E(1 + i\tan\delta)$ can be experimentally derived in vitro for muscle, between 40 Hz and 120 Hz[38]. Whilst the trend is somewhat linear within this frequency range, extrapolating throughout the audio range would yield unphysical values at higher frequencies. We use $\tan\delta=0.3$, which corresponds to the value measured in human muscle at 100 Hz[38]. For amniotic fluid, we use the properties of water with an attenuation coefficient obtained at 37 °C. For soft tissue and bone, we assume a linear power absorption law with frequency. For the amniotic fluid, we assume that the medium attenuation is frequency-squared dependent,

as is the case for water. As a result of the variability and patient specificity of the speed of compressional waves and density for soft tissue and bone, we use values consistent with those in the literature[39,40].

## Computational protocol

The open-source Python library OptimUS v0.2.1[35] was used to simulate sound pressure levels in the entire computational domain and in 12th octave bands between 20 Hz and 20 kHz resulting in a total of 128 frequencies. The simulations in the present study were performed on a desktop machine (Dell Precision 32 core, 512 GB RAM). Hierarchical matrix compression techniques[41] and dedicated preconditioners[32] significantly reduce the memory footprint and increase the convergence rate of iterative solvers. Acoustic transmission problems across high-contrast media can be efficiently and accurately solved for high $ka$ scenarios[36], where $k$ is the wavenumber and $a$ the dimension of the scatterer. This product is of significance in acoustics as it represents a dimensionless quantity that relates the wavelength to the physical dimension of the domain. A distinct advantage of the BEM is that it suffers only minimal numerical dispersion and pollution[42] effects. Numerical dispersion arises in finite-difference time domain schemes as well as k-space pseudospectral methods when the phase velocity of numerical wave modes differs from its true value by an amount varying with the wavelength, direction of propagation in the grid, and grid discretisation[43]. As a result of this artifact, propagating numerical waves accumulate delay or phase errors that can lead to nonphysical results. Numerical pollution effects occur when, as $k \to \infty$, the total number of degrees of freedom required to maintain computational accuracy grows faster than $k^n$, where $n$ is the dimension of the physical domain in which the problem is formulated[42]. Another advantage of the BEM is that domain truncation effects are not a concern due to the imposition of the Sommerfeld radiation condition at infinity. Acoustic pressures at degrees of freedom on the surface meshes are initially obtained and field SPLs were inferred using the appropriate potential operators[32] and the use of triangular surface meshes avoids unwanted staircasing effects.

## Acoustic transmission in utero

The frequency response in 12th octave bands between 20 Hz and 20 kHz was calculated, using 1000 Hz as the reference middle frequency. Instead of focusing on locations inside the uterus specific to the fetus' morphology (e.g., ears or head), we opted to evaluate the acoustic pressures throughout the whole uterus. Indeed, the fetus is not static inside the womb throughout the gestational period. Whilst most fetuses are in the head-down position, they may be in breech, or transverse position. Furthermore, general movement and activity of the mother which includes pose change and respiration will also result in the fetus being displaced within the womb. We require a metric that will provide a spatial average of acoustic pressure quantities inside the uterus. If we were to consider the complex acoustic pressure and produce a spatial average of this quantity, we may be underestimating the transmission of external sound sources, due to destructive interferences owing to the inclusion of phase information. We therefore instead consider the metric described in Section 3.7 of ISO 10052:2021[44] known as the impact SPL. This is effectively obtained from a spatial root mean square (RMS) of the pressure magnitudes, where we used calculated acoustic pressure values along a 3D Cartesian grid of points inside the uterus. The impact SPL is closely related to the $\ell^2$-norm of the acoustic pressure in the uterus. It is obtained, in dB scale, as follows:

$$L_{\text{uterus, RMS}} = 20 \log_{10} \sqrt{\frac{1}{N} \sum_{i=1}^{N} |p_i|^2} \qquad (7)$$

where $N$ is the total number of grid points considered in the uterus, which is discussed in the Methods Section, and $p_i$ represents the spatial component of the acoustic pressure at the $i$th location.

The RMS metric can be interpreted as the average noise exposure level of the fetus. However, depending on the positioning, the fetus may be exposed to local peaks due to modal acoustics in the abdomen. Hence, in addition to $L_{\text{uterus,RMS}}$, the $\ell^\infty$-norm was also evaluated, which is effectively the maximum value of the acoustic pressure magnitude inside the uterus evaluated across the sample points. In dB scale, this quantity is given by:

$$L_{\text{uterus}, \ell^\infty} = 20 \log_{10} \max |p|, \qquad (8)$$

Finally, the acoustic pressure at the barycentre of the uterus was calculated, also as a function of frequency throughout the human audio range in 12th octave bands. This provides a point measurement for which magnitude and phase information will be used to derive the filters used for convolution with audio signals described in the Methods Section. The frequency responses for these three pressure quantities are displayed for the four datasets in Fig. 2A−L.

The plots in Fig. 2A−L exhibit a range of common features. It can first be noted that between 20 Hz and 1 kHz, the attenuation in the RMS and barycentre calculations is within −6 dB of the amplitude of the incident wave, indicating that the systems under consideration exhibit a quasi-flat frequency response within this passband for compressional waves. Furthermore, additional calculations indicate that this extends to infrasound frequencies, i.e., below 20 Hz down to 0 Hz. 1 kHz falls around the midrange of human hearing and is just below the fundamental frequency of a B5 on a musical instrument (987.77 Hz). On a guitar in standard tuning, this corresponds to the 19th fret on the high E string and is just one semitone below the soprano high C, C6 (1046.502 Hz)[45]. Hence, for the transmission of compressional waves in utero, these simulations suggest that the developing fetus is exposed to noises that are virtually unattenuated below 1 kHz, regardless of the acoustic pressure quantity investigated (RMS, $\ell^\infty$-norm or sampled at a specified point) and the acoustic attenuation coefficient considered for the uterus. This frequency range encompasses much of the human speech spectrum (~300–3000 Hz)[46–48] as well as musical sounds. Low-frequency noises, such as those encountered in urban environments and in occupational noise settings are likely to be fully transmitted. This will include portions of the spectrum of noise sources such as road vehicles, aircraft, industrial machinery, artillery and mining explosions, as well as air movement machinery such as wind turbines, compressors, and ventilation or air-conditioning units[49].

Despite the common traits shown in all four datasets across the acoustic pressure quantities investigated, there exist important distinctions. The SPL of the spatial RMS of the acoustic pressure magnitude will tend to overestimate the transmitted acoustic pressure, as it is effectively the result of a spatial root mean square of the pressure magnitude at designated regularly spaced locations across the uterus. This quantity is nevertheless useful for assessing the potential for resonant behavior within this region, which is visible in Fig. 2A, D, G, J, in the form of local maxima at frequencies above 3 kHz, in the case of lower attenuation inside the uterus. This confirms the results of prior in vivo studies[29,31,50] as well as an experimental study involving acoustic transmission into a non-invasive assessment of acoustic fields acting on the fetus, which employed a soft capsule filled with liquid[51], and which showed that transmission of waves up to 1 kHz was unaffected by the configuration. We note that modal behavior associated with the dataset at the earliest stage of gestation considered in this study (Subject 1–25 weeks and 1 day) occurs above 7 kHz in the low-attenuation case (see Fig. 2A). For the dataset with the latest gestational age considered in this paper (Subject 4–37 weeks and 3 days), this occurs above 6 kHz, therefore at a comparatively lower frequency. Broadly, it is expected that smaller anatomical dimensions will lead to resonances occurring at higher frequencies. Furthermore, at frequencies above 3 kHz, in the case of the uterus featuring a lower attenuation coefficient, it can be noted that the magnitude of the

transmitted wave can at certain frequencies exceed that of the incident wave, effectively amplifying the signal due to reflections and acoustic modes. This is the case for Subject 2 and Subject 3 datasets in the low-attenuation scenario in the uterus and in the case of the acoustic pressure being sampled at the barycentre of the uterus, as shown in Figs. 2C, 4C. Numerical experiments on spheres scattered by plane

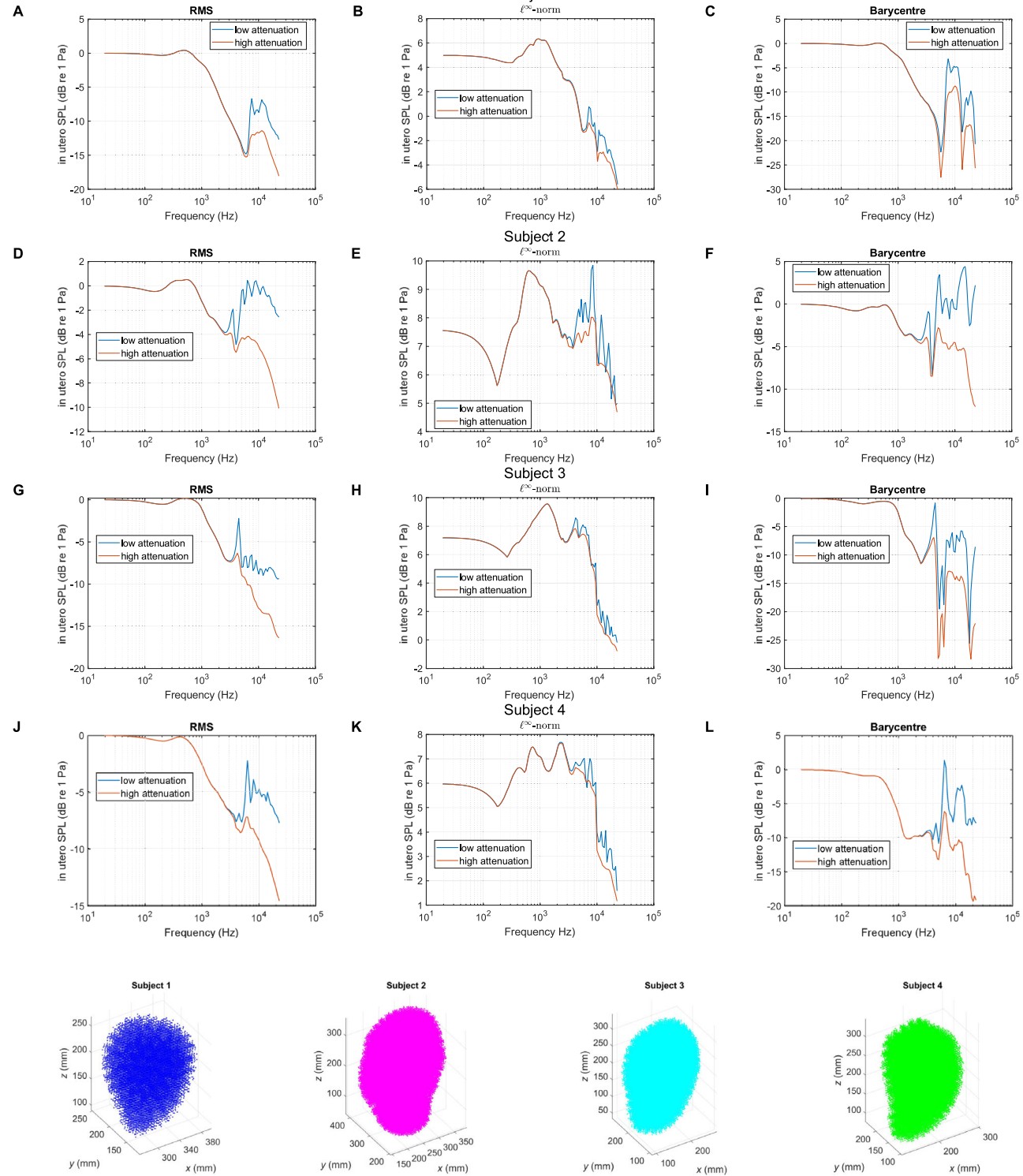

**Fig. 2 | Sound pressure level inside the uterus as a function of frequency for a unit amplitude incident plane wave.** Frequency response plots of the sound pressure level (SPL) inside the womb were obtained for a unit amplitude plane wave traveling towards the front of the body, in the negative $x$ direction. Such a plane wave is described mathematically by the real part of $e^{-i(\omega t - kx)}$ where $k$ is the wave number in air and $\omega$ is the angular frequency. Three metrics of the sound pressure level inside the uterus are plotted for datasets associated with Subjects 1, 2, 3 and 4. **A**, **D**, **G** and **J** correspond to the SPL resulting from the spatial RMS of the acoustic pressure magnitude inside the uterus; **B**, **E**, **H** and **K** describe the SPL associated with the $\ell^\infty$-norm, effectively corresponding to the maximum pressure magnitude at the sampled points; **C**, **F**, **I** and **L** represent the SPL resulting from the acoustic pressure magnitude at the barycentre of the uterus. Uterus points within a solid angle of 0.5 steradian from the surface of the mesh were discarded in the analysis as the BEM can overestimate field potentials close to a surface. The field potential evaluation points for subjects 1, 2, 3, and 4 are displayed below in blue, magenta, cyan, and green, respectively.

waves using our model also display these findings[35] and have been validated with the known analytical solution[52]. For the calculations employing the higher attenuation coefficient in the uterus in Fig. 2A, D, G, J, the resonances are dampened, as expected, but the transmission remains within 15 dB of the incident wave across the human audio range in datasets associated with Subjects 2 and 3 when considering the acoustic pressure magnitude sampled at the uterus barycentre. These calculations of acoustic quantities at a specific point provide a representation of local effects inside the uterus. The $l^\infty$-norm plotted in Fig. 2B, E, H represents worst-case scenarios, whereby the maximum SPL transmitted inside the uterus is plotted as a function of frequencies throughout the human audio range. It should be noted that the locations at which these maxima occur will vary with frequency. We note that for all datasets, in the cases of both low and high acoustic attenuation inside the uterus, the SPL associated with the $l^\infty$-norm is virtually always greater than 0 dB. At the midrange frequency of 1 kHz, we note that the transmitted sound pressure level is 9 dB above that of the incident wave for datasets associated with Subjects 2 and 3, 7 dB above for Subject 4, and 6 dB for Subject 1. This is due to multiple reflections which occur inside the maternal abdomen and other anatomical groups, and which constructively combine at specific locations to amplify the acoustic pressure magnitude associated with the incident wave.

To contrast this data with that obtained from experiments on ovine models[31], we note that the simulations in this paper correspond to a free field environment, i.e., in an unbounded domain where the Sommerfeld radiation condition at infinity applies. The experiments on ovine models took place in an operating theater[31], which included a highly reverberant environment, therefore providing an overestimate of the incident acoustic field and with the measured transfer characteristics including the room impulse response. This, therefore, resulted in a low frequency response below 0 dB. Otherwise, we observe similar trends in terms of the decrease in the transfer characteristics at frequencies above 1 kHz.

### Visualization of in utero sound transmission
Figures 3, 4, 5, and 6 show the SPL transmitted in utero at frequencies of 5, 10, and 20 kHz, for a unit amplitude plane wave incident onto the maternal abdomen of Subjects 1, 2, 3 and 4, respectively. The plane of visualization is the transverse plane at the midpoint of the height of the uterus along the Cartesian z-axis. For each dataset, two different acoustic attenuation coefficients are used for the uterus, as described in Table 1: that of amniotic fluid in the low attenuation case and that of muscle tissue in the high attenuation case.

In Figs. 3–6, we note that the incident plane wave traveling along the negative x direction is reflected at the air/soft tissue interface at the abdomen and that the incident wave and scattered waves interact constructively and destructively with one another, generating interference patterns. We note the presence of a shadow zone behind the lower back area. These maps allow for the intricacies and complexities of the acoustic pressure fields to be appreciated. Indeed, whilst the data in Fig. 2 demonstrate the extent of in utero sound transmission, the pressure maps in Fig. 3–6 establish the increase in modal and standing wave patterns at frequencies above 5 kHz, where the wavelength in soft tissue is around 30 cm, which is of the order of the abdominal region. In particular, modal behavior inside the uterus is observed in Fig. 3E, K, at 10 kHz and Fig. 5D, J, at 5 kHz. Also, we note the presence of an interference pattern in the uterus of Subject 4 in Fig. 4F, L in the lower acoustic attenuation case.

### Transmission of sound sources inside the womb: convolution with audio signals
With a view of providing an impression of in-utero acoustic transmission, a reference soundscape was generated from a range of audio signals that feature, in chronological sequence:

- A London Underground train leaving and arriving at a station[53]
- A segment of an instrumental ambient rock music composition[54]
- Ambient crow noise obtained from the Louvre museum[53]
- Crowd applause[53].

A causal, linear and time-invariant filter was obtained as outlined in the Methods section based on in utero calculations on datasets associated with Subjects 2 and 3, for the pressure at the barycentre of the uterus using the attenuation coefficient of uterine tissue (high attenuation case). The reference soundscape was convolved with this filter to yield an impression of in utero sound transmission. The reference and filtered soundscape audio filenames are:

- Reference unfiltered soundscape: Supplementary_Audio_1.MP3
- Subject 2: Supplementary_Audio_2.MP3
- Subject 3: Supplementary_Audio_3.MP3

To appreciate the subtleties introduced by the filtering, it is recommended that the soundscapes be listened to on good-quality headphones and/or a high-fidelity sound reproduction system.

## Discussion
Using a computational method based on state-of-the-art BEM formulations, we found that the human pregnant abdomen permits significant spectral content through to the uterus and that content below 1 kHz, is attenuated by as little as 6 dB. This finding was consistent for all datasets and acoustic pressure metrics evaluated and is in agreement with in vivo data obtained in prior studies[29,31] on pregnant ovine models. Our study also shows how detailed acoustic pressure maps for external sound sources can be displayed, showcasing the complexities of the fetal auditory environment.

Our methodology made some simplifications in the design of our mathematical model, which may impact the final results. Indeed, not all anatomical groups have been considered as we have constrained our analyses to include only the maternal abdomen, the uterus and the maternal spine. Given that the wavelength in soft tissue does not fall below 7 cm at 20 kHz, finer anatomical detail is unlikely to produce additional crucial information for the evaluation of sound fields in utero. In addition, MRI scans were focused on obtaining imaging from the uterus, placenta, and fetus rather than the whole maternal abdomen, thus the upper part of the maternal abdomen was not included in the acoustic propagation path for our simulations. Inclusion of the upper abdomen would have required moving the patient's position to optimize data acquisition with additional time spent undergoing imaging which was not possible during one imaging session. However, the truncations along the two transverse planes may not have an impact on the predicted acoustics pressures as the anatomical regions above the uterus is made mainly of the lungs, which are air-filled, and which will reflect acoustic waves in a manner not unlike the soft tissue/ air interface in the upper transverse plane truncation in our computational mesh. Similarly, the lower transverse plane, which features the legs, will also feature an additional soft tissue/ air boundary.

We have investigated late third-trimester scenarios, between 32 and 37 weeks. It is expected that in utero transmission of external sound sources will be affected as a function of gestational age due to changes in the morphology of the maternal abdomen, and the fetal position as well as changes in amniotic fluid volume as pregnancy advances.

Incomplete knowledge of acoustic attenuation coefficients for compressional waves in soft tissue and bone at audio range frequencies is also a possible source of uncertainty in the computations. We have extrapolated low-frequency data for muscle tissue and assumed an absorption power-law frequency dependence. To mitigate this assumption, we have presented two extremes of possible attenuation scenarios, corresponding to that of amniotic fluid and muscle tissue, respectively. The

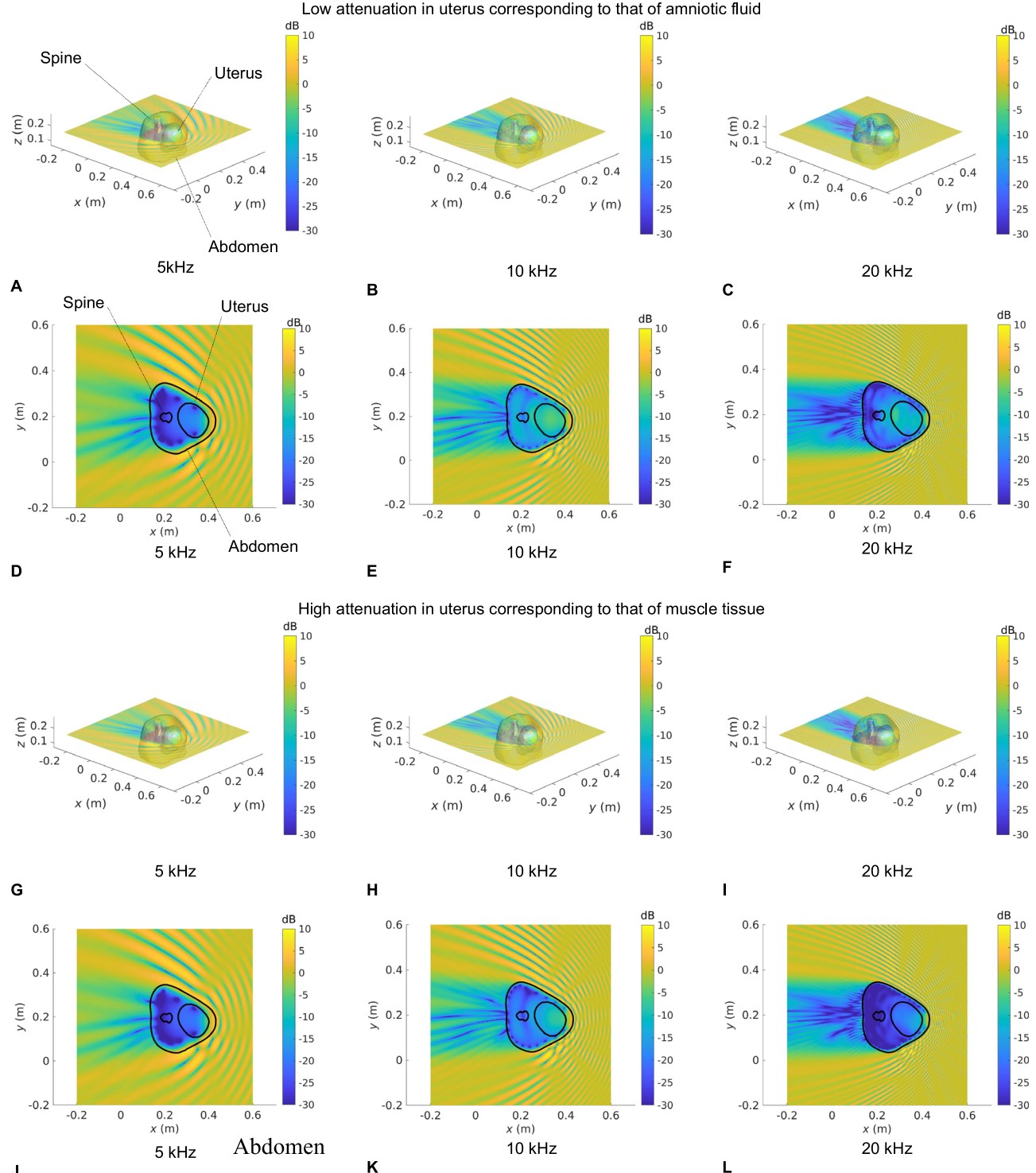

**Fig. 3 | Sound pressure level maps at 5, 10, and 20 kHz for Subject 1 for an incident unit amplitude plane wave.** SPL inside all anatomical regions for an incident unit amplitude plane wave traveling in the negative *x* direction. The acoustic attenuation coefficient in the uterus is that of amniotic fluid in **A**–**F** and that of muscle tissue in **G** – **L**. 3D maps of the SPL re 1 Pa are shown in **A**–**C** and **G**–**I**. **D**–**F** and **J**–**L** show a slice of the SPL re 1 Pa in the transverse plane passing through the barycentre of the uterus. Anatomical groups and contours are labeled in **A** and **D**, respectively.

characterization of the nature of damping mechanisms at audio-range frequencies in soft tissue and bone, as well as the identification of a suitable damping model and its relationship with frequency, requires further studies.

Our analysis was focused on the propagation of compressional waves in the body, treating the tissue groups as acoustic media. It is well known that both soft tissue and bone support the propagation of shear waves[55,56]. Shear wave mode conversion could in principle occur, depending on the incident acoustic field, adding more complexity to the problem of in-utero sound transmission and propagation. This limitation could be addressed by using a viscoelastic boundary element formulation. However, this would be more computationally demanding owing to the increase in the number of degrees of freedom resulting from having to solve for vector quantities and from the

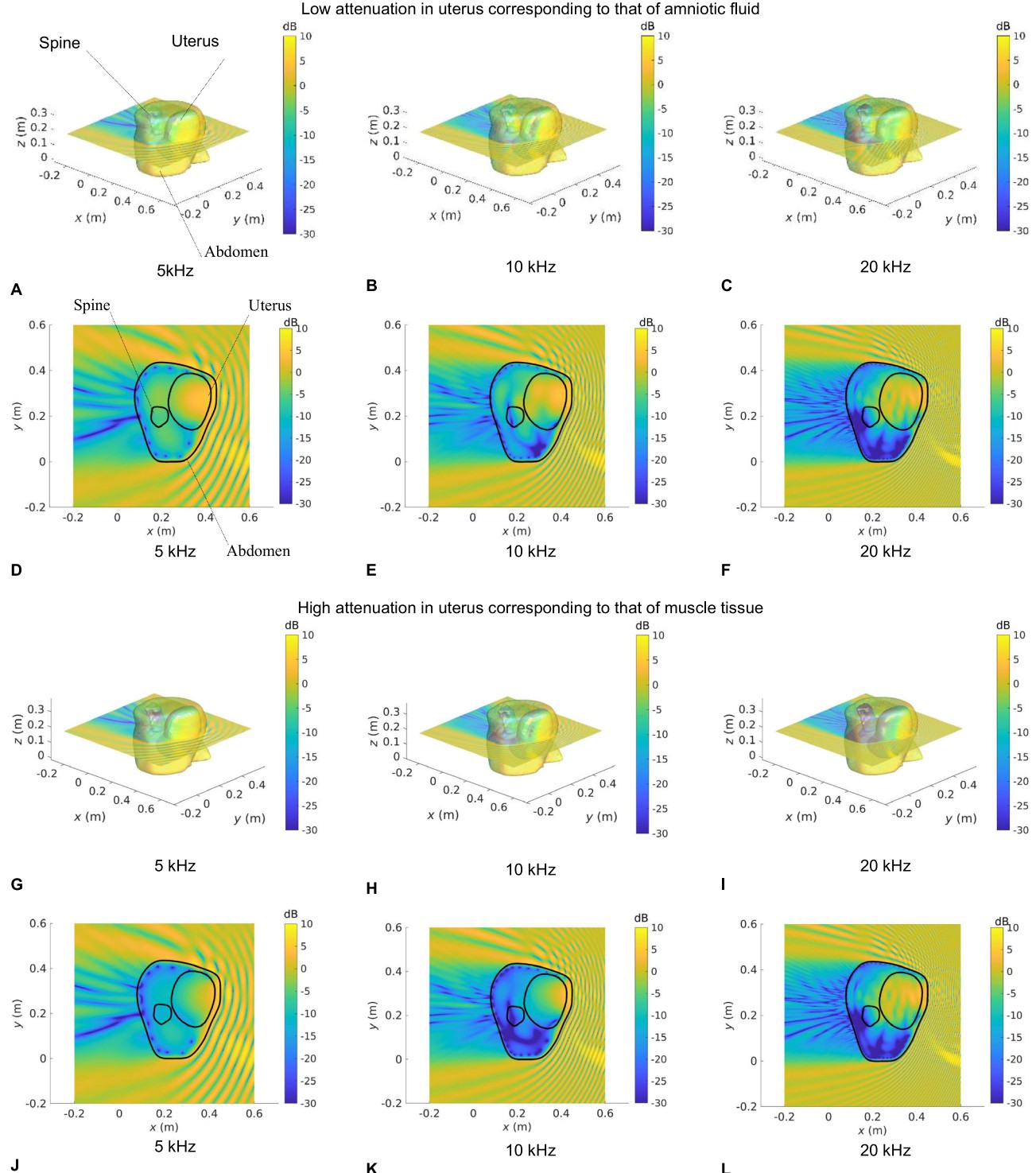

**Fig. 4 | Sound pressure level maps at 5, 10, and 20 kHz for Subject 2 for an incident unit amplitude plane wave.** SPL inside all anatomical regions for an incident unit amplitude plane wave traveling in the negative $x$ direction. The acoustic attenuation coefficient in the uterus is that of amniotic fluid in (**A–F**) and that of muscle tissue in (**G–L**). 3D maps of the SPL re 1 Pa are shown in (**A–C**) and (**G–I**). **D–F** and **J–L** show a slice of the SPL re 1 Pa in the transverse plane passing through the barycentre of the uterus. Anatomical groups and contours are labeled in (**A**) and (**D**), respectively.

---

denser meshes which would be required to resolve the shorter wavelengths associated with shear waves.

We aimed to better understand the fetal exposure to exterior noise sources and have limited our analyses to acoustic plane waves as the incident exterior sound field. The developing fetus is also exposed to physiological sounds as well as the transmission of the maternal voice via anatomical paths, mainly by bone conduction. Our model can

be extended to include any number of monopole and dipole sources, as well as combinations of Neuman and Dirichlet source boundary conditions, all of which will affect the interior sound field.

Our current study on sound transmission in utero has several strengths which altogether have the merit of addressing important features which potentially have significant ramifications for fetal neurobiological development. We have developed a validated

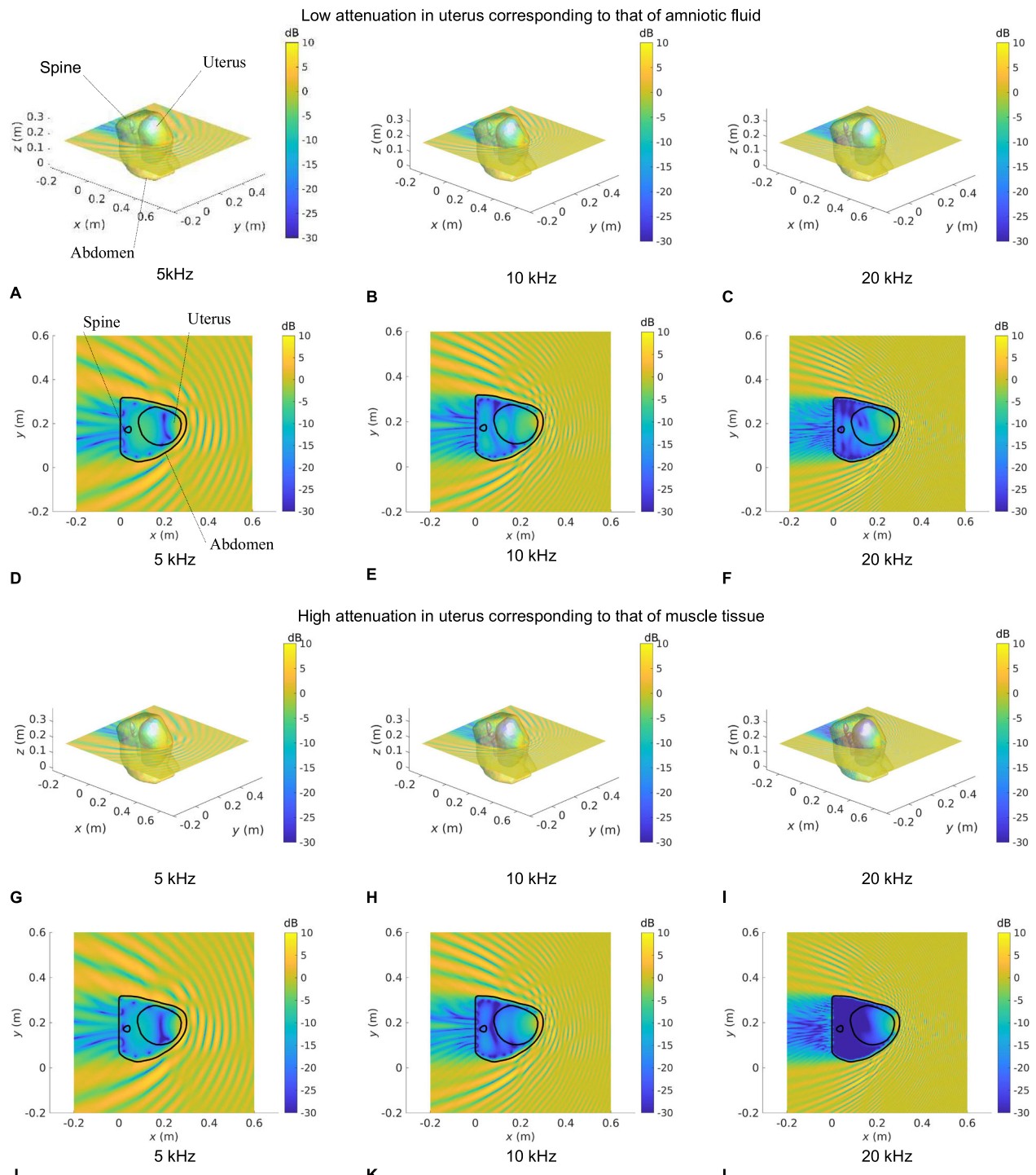

**Fig. 5 | Sound pressure level maps at 5, 10, and 20 kHz for Subject 3 for an incident unit amplitude plane wave.** SPL inside all anatomical regions for an incident unit amplitude plane wave traveling in the negative *x* direction. The acoustic attenuation coefficient in the uterus is that of amniotic fluid in **A–F** and that of muscle tissue in **G–L**. 3D maps of the SPL re 1 Pa are shown in **A–C** and **G–I**. **D–F** and **J–L** show a slice of the SPL re 1 Pa in the transverse plane passing through the barycentre of the uterus. Anatomical groups and contours are labeled in (**A**) and (**D**), respectively.

computational model capable of predicting acoustic pressure transmission at high frequencies relative to the wavelength and in high-contrast scenarios. A prior attempt was made to carry out this type of analysis with the finite element method (FEM) which demonstrated the inapplicability of the technique throughout the whole of the human audio range, owing to numerical pollutions effects[57] reaffirming the validity and superiority of our approach. We expect that the results discussed in this paper will provide a scientific starting point to

establish noise dose and exposure safety levels for the developing fetus at various stages of gestation. This will include occupation noise, leisure noise, urban noise as well as noise resulting from medical diagnostic interventions such as MRI scans[58]. Numerical simulations on anatomical data may also be used as a predictor of the risk of specific noise profiles on the developing fetus. For example, if there is a significant overlap of the spectral content of the noise close to or at a known resonance, this could potentially enhance mechanical stresses

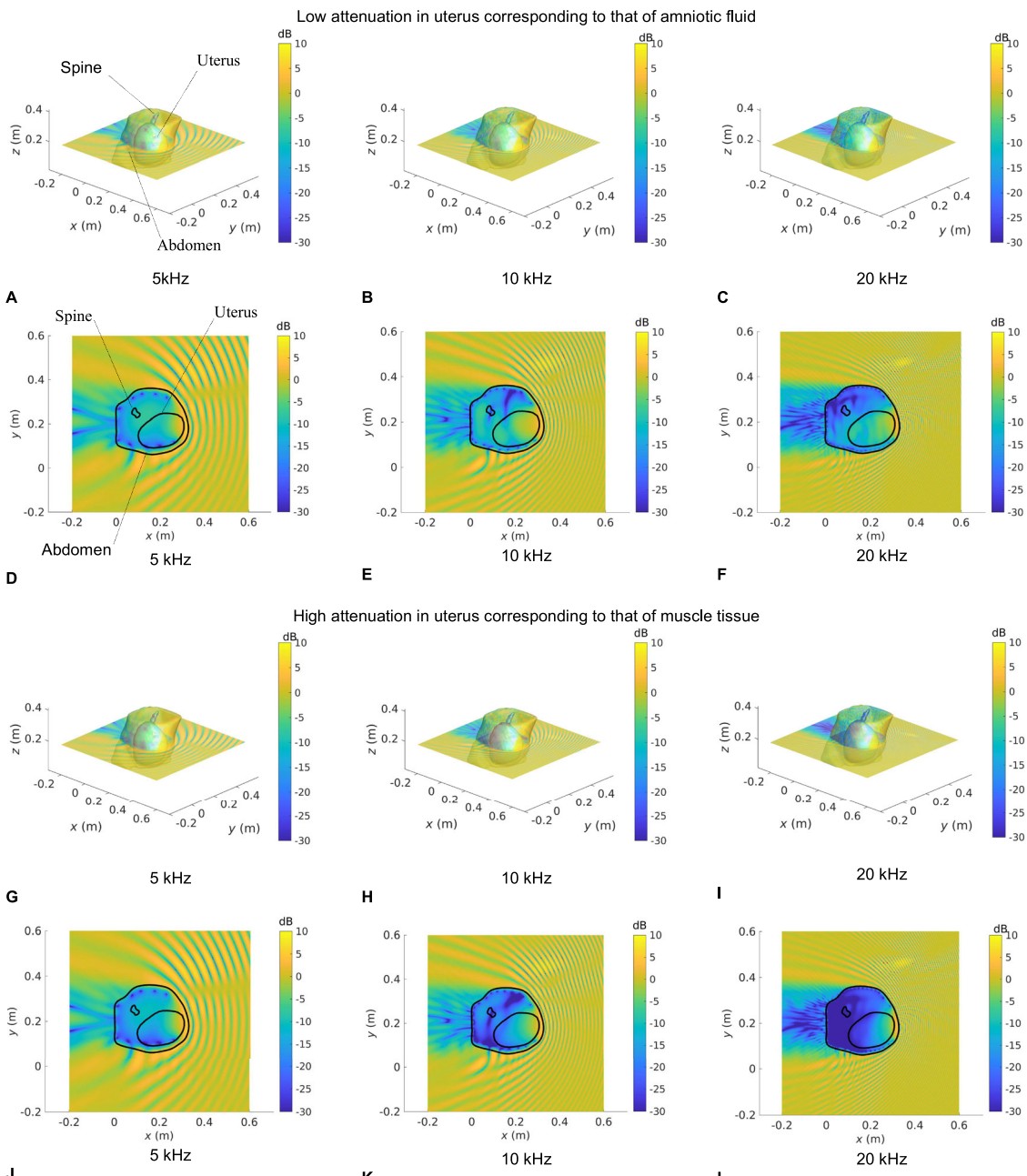

**Fig. 6 | Sound pressure level maps at 5, 10 and 20 kHz for Subject 4 for an incident unit amplitude plane wave.** SPL inside all anatomical regions for an incident unit amplitude plane wave traveling in the negative $x$ direction. The acoustic attenuation coefficient in the uterus is that of amniotic fluid in (**A–F**) and that of muscle tissue in (**G–L**). 3D maps of the SPL re 1 Pa are shown in (**A–C**) and (**G–I**). **D–F** and **J–L** show a slice of the SPL re 1 Pa in the transverse plane passing through the barycentre of the uterus. Anatomical groups and contours are labeled in (**A**) and (**D**), respectively.

## Table 1 | Acoustic properties of anatomical groups used as input data in OptimUS calculations

| Anatomical group | Speed of sound (m·s⁻¹) | Material density (kg·m⁻³) | Attenuation coefficient (Np·m⁻¹ at 1 MHz) | Attenuation power law |
|---|---|---|---|---|
| Abdominal tissue | 1489 | 950 | 0.1 | 1 |
| Spine bone | 4020 | 2700 | 0.2 | 1 |
| Uterine tissue | 1500 | 1000 | 585.3 | 1 |
| Amniotic fluid | 1500 | 1000 | 15·10⁻³ | 2 |

inside the uterus. Patient-specific risk would also be dependent on tissue acoustic properties and heterogeneity. Developing dedicated uncertainty analyses may allow for establishing the risk profile of an individual subject to specific external noise sources, at a given gestational age.

## Methods
### Ethics approval
This study complies with all relevant ethical regulations for research involving human participants. The study involving MRI protocols was approved by the UK National Research Ethics Service and all participants gave written informed consent (London – Hampstead Research Ethics Committee, REC reference 15/LO/1488).

**Table 2 | Degrees of freedom of computational meshes for each dataset in the frequency range subgroups considered**

| Dataset | $f$<2.5 kHz | 2.5 kHz < $f$≤5 kHz | 5 kHz < $f$≤10 kHz | 10 kHz < $f$≤15 kHz | $f$>15 kHz |
|---|---|---|---|---|---|
| Subject 1 | 1235 | 3131 | 6352 | 9026 | 18,251 |
| Subject 2 | 3111 | 7307 | 16,576 | 20,557 | 41,739 |
| Subject 3 | 1778 | 4731 | 9696 | 14,148 | 29,345 |
| Subject 4 | 1708 | 4286 | 8599 | 12,409 | 24,885 |

## Computational meshes

Meshes for the datasets associated with Subjects 1, 2, 3, and 4, were generated in Autodesk Meshmixer v3.5[59], following the segmentation of MRI scans. The three anatomical groups meshed were the lower abdominal region, the spine, and the uterus. The datasets were smoothed and patched to obtain closed surfaces. Three-noded triangular elements were used and a strategy was adopted whereby the mesh density of each geometric group was varied as a function of the frequency of excitation and the acoustic properties of the media on both sides of the interface. For example, for the abdominal region, which is in contact with air, the external medium, we use a mesh density determined by the wavelength in soft tissue at the frequency of excitation. The meshes were then converted to Gmsh v4.13.1[60] format for reading in OptimUS. Based on convergence tests carried available on the OptimUS repository[35], 4 to 5 triangular elements per wavelength are sufficient to ensure convergence of the Generalized Minimal Residual Method (GMRES) solver and generate results within 7.5% of the analytical solution on nested spheres[52]. To reduce run times associated with the frequency sweep calculations, we divide the audio range 12th octave band frequency spectrum into five subgroups, as shown in Table 2, where the resulting number of degrees of freedom is displayed.

## MRI patient data segmentation

MRI examinations were performed on a 1.5 T magnet (MAGENETOM Avanto; Siemens Healthcare). Four subjects of female biological sex beyond 24 weeks of gestational age (confirmed by dating scan) with uncomplicated pregnancies had MRI data acquired. No compensation was awarded to the participants and all were volunteers. The ages of the patients at the time of scanning were as follows:

- Subject 1: 32 years of age
- Subject 2: 34 years of age
- Subject 3: 36 years of age
- Subject 4: 36 years of age.

Patients were placed in the left lateral decubitus position and had a moderately filled bladder. The uterus was imaged in at least 3 orthogonal planes (axial, coronal, and sagittal) relative to the placenta-myometrium interface. The protocol consisted of T2-weighted fast acquisition spin echo sequences (HASTE). The following parameters were applied: slice thickness (4 mm), spacing between slices (4 mm), repetition time (1013.8 ms), echo time (89.6 ms), flip angle (107.9°), and pixel spacing (0.74–0.74 mm).

Imaging data was manually segmented using ITK-SNAP to provide multiple tissue segmentations for the maternal abdomen, uterus, fetal body and brain, placenta, and maternal spine.

## Boundary element model

The Helmholtz equation is commonly used for modeling harmonic wave propagation in acoustic phenomena like room acoustics, sonar, and biomedical ultrasound[61]. Among numerical methods, the boundary element method (BEM) stands out as an efficient approach for solving Helmholtz transmission problems[33,34,36]. Unlike the FEM and other volumetric solvers, which directly discretize partial differential equations within the volume of interest, BEM first transforms the equations into a boundary integral formulation. This formulation depends on the geometry of the problem, specifically the interfaces between volumetric subdomains with constant material parameters (e.g., density and speed of sound). The volumetric partial differential equations are rewritten into a representation of the acoustic fields in terms of surface potentials at the material interfaces. Scientific literature provides various boundary integral formulations tailored to specific geometries, including single scatterers, multiple disjoint scatterers, and nested domains[62,63]. In this study, we employ a dedicated formulation designed for piecewise homogeneous domains, allowing for efficient simulations by combining different types of boundary integral formulations. The specific nested domain formulation applied to the topologies specific to this paper used is detailed in Supplementary Information section. The formulation has been generalized to include arbitrary combinations of disjointed multiple scatterers and nested domains[35]. This design process simplifies the generalization of the BEM to more complex geometries and allows for efficient simulations by combining different types of boundary integral formulations.

The BEM employed in this paper assumes a Helmholtz transmission problem and uses a combination of multiple-domain Poggio-Miller-Chan-Harrington-Wu-Tsai equations and on-surface radiation condition (OSRC) preconditioners[32]. This formulation is described in Supplementary Information section for the specific case of a bounded domain embedded in free space with two other domains inside. The damped wavenumber in the OSRC preconditioner is set to: $\lambda_{min} + 0.4\,i\,\lambda_{min}^{-\frac{1}{3}}0.001^{-\frac{2}{3}}$ where $\lambda_{min}$ corresponds to the smallest wavenumber of the media considered, in this case air.

We used hierarchical matrix compression to reduce the problem size with the parameters set as follows:

- $\epsilon = 10^{-6}$
- maximum rank = 1000
- maximum block size = $10^6$

We considered the excitation acoustic wave to be a unit amplitude plane wave incident on the maternal abdomen. At each frequency, we calculate the Neumann and Dirichlet traces at the surface of the anatomical regions. The GMRES solver, without restart, converged in all cases to a tolerance of $10^{-4}$ in the residual norm, within less than 2000 iterations.

We then calculated the spatial RMS of the acoustic pressure magnitude inside the volume of the uterus, as well as the $\ell^{\infty}$-norm of the pressure magnitude within this region, followed by the magnitude of the acoustic pressure at its barycentre.

To add an additional layer of validation to this approach, we considered two concentric spheres with dimensions similar to the abdominal region and uterus in our datasets. The outer sphere has diameter of 0.5 m and the inner sphere of 0.3 m. We substituted anatomical computational grids for those representing these spheres and we carried out the 12th octave band frequency sweeps using the above protocols and compared the results with the analytical solution[52] for an incident plane wave traveling in the positive $x$ direction. We calculate the SPL resulting from the spatial RMS of the acoustic pressure magnitude inside the inner sphere as well as the acoustic pressure magnitude at the barycentre of the inner sphere. The outer sphere was assigned the properties of abdominal tissue and the inner sphere those of the uterus, for both the high (muscle tissue) and low (amniotic fluid)

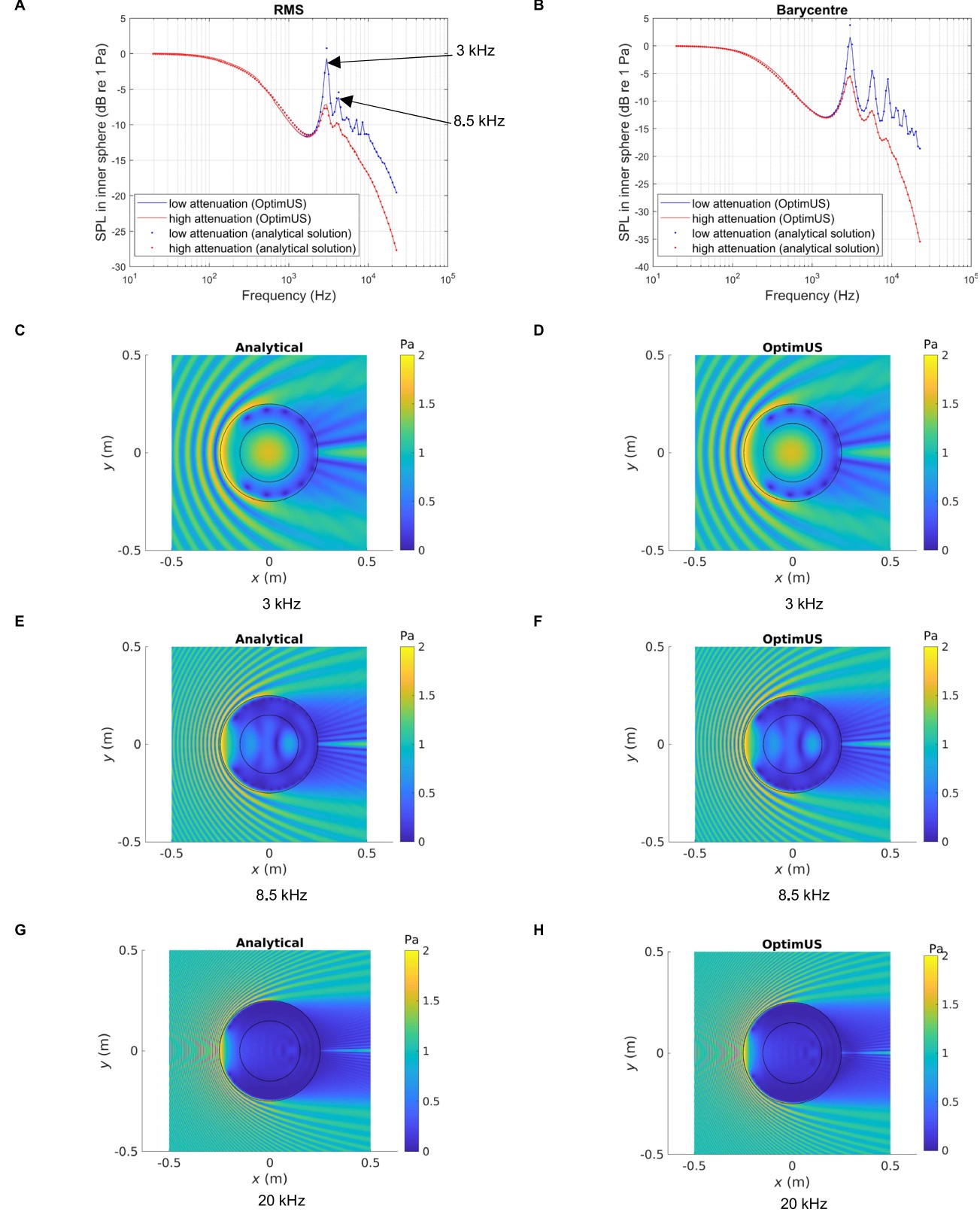

attenuation cases. The results, shown in Fig. 7A, B, demonstrate agreement generally within ±0.5 dB between the BEM and the analytical solution, with the exception of resonances and antiresonances, where slight differences between numerical and analytical solutions may occur. We highlight two resonances at 3.5 kHz and 8 kHz, which are consistent with the results on anatomical data. Further validation is described in Fig. 7C–H where the acoustic pressure magnitude is

plotted at the two resonant frequencies denoted above, as well as 20 kHz, the upper limit of the human audio range. We note agreement with the analytical solution.

## Convolution with audio signals
Based on the magnitude and phase of the predicted acoustic pressure obtained at the barycentre of the uterus for each dataset, filters from

**Fig. 7 | Validation of OptimUS computational model against the analytical solution for nested spheres: frequency response and acoustic pressure field visualization.** Sound pressure level transmission inside the inner sphere with dimensions representative of the uterus as a function of frequency for a unit amplitude incident plane wave traveling in the positive $x$ direction for two concentric spheres, of radii 0.25 m and 0.15 m, obtained from **A** the SPL resulting from the spatial RMS of the acoustic pressure magnitude inside the inner sphere with two resonances shown at 3 kHz and 8 kHz, and **B** the acoustic pressure magnitude

at the center of the inner sphere. The exterior medium is air. The medium bounded by the exterior domain and the inner sphere has the properties of abdominal tissue and the inner sphere those of amniotic fluid. Acoustic pressure magnitude in the Cartesian $x - y$ plane, describing the interactions between the incident plane wave and the concentric spheres at excitation frequencies of 3 kHz, 8.5 kHz, and 20 kHz is shown in (**C**, **E** and **G**) which are obtained using the analytical solution[52]. **D**, **F**, and **H** correspond to the nested BEM solution provided by OptimUS.

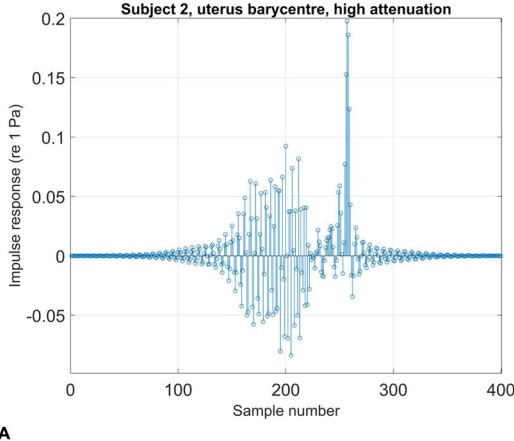

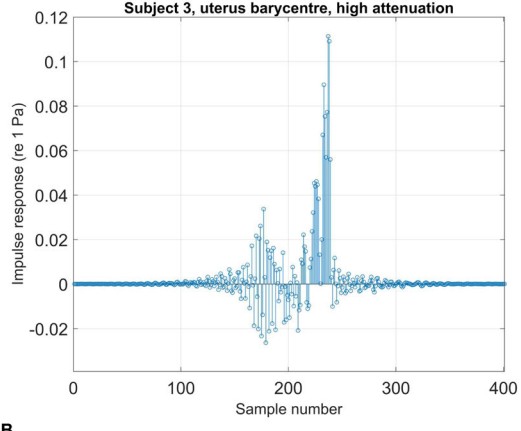

**Fig. 8 | Impulse response generated from the acoustic pressure inside the uterus.** Impulse response generated from the complex acoustic pressure at **A**, the barycentre of the uterus of Subject 2 and **B**, the barycentre of the uterus of

Subject 3, in response to a unit amplitude plane wave incident on the abdomen traveling in the negative $x$ direction.

the datasets corresponding to Subjects 2 and 3, were obtained using spline interpolated data on the magnitude and unwrapped phase responses. The attenuation coefficient of uterine tissue is used to describe the uterus, thereby corresponding to the high attenuation case. A sampling frequency of 44.1 kHz was assumed, and 16,385 interpolation points were used. A constant delay of 200 samples at each frequency was introduced to linearize the phase. Then, a finite impulse response (FIR) filter was estimated using the least-square method. The order of the filter was increased to minimize the mean-square error between the frequency response function of the filter and the predicted spectrum of the acoustic pressure. The impulse response of this FIR filter was then used to convolve a set of audio signals, including hand clapping, crowd noise, the London Underground and rock music. The signal processing was carried out using MATLAB R2024b (MathWorks, 2024). The filter impulse responses are shown in Fig. 8.

### Reporting summary
Further information on research design is available in the Nature Portfolio Reporting Summary linked to this article.

## Data availability
The source data generated in this study are provided in the Source Data file, available at https://zenodo.org/records/15052299.

## Code availability
The code in this paper, along with sample Jupyter notebooks is available from the GitHub repository https://github.com/optimuslib, which is also available at Zenodo. The sample notebook demonstrating single-frequency simulations in this paper can be found at this link. Any additional scripts used to generate or visualize the results are available upon reasonable request to the corresponding author, for research and educational purposes only, with a timeframe of response of two weeks.

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

## Acknowledgements

The authors wish to thank Sonic Womb Productions Ltd and Nathalie Samani, Ph.D., for their technical support; ALD is supported by funds from the National Institute for Health Research University College London Hospitals Biomedical Research Center. This work was also supported by a grant entitled "Optimizing patient-specific treatment plans for ultrasound ablative therapies in the abdomen (OptimUS)" from the Engineering and Physical Sciences Research Council (EPSRC) (Grant Nos. EP/P013309/1 to the University of Cambridge and EP/P012434/1 to University College London). EW is supported by FONDECYT 1230642, ANID, Chile. A.M. and A.L.D. were supported by the MRC (MR/X010007/1), the NIH (R01 HD108833), the BBSRC (BB/Y514214/1), and EPSRC (NS/A000027/1).

## Author contributions

Conceptualisation: A.T.M., E.J., J.H., P.G.; Data acquisition: N.M., A.M., A.L.D., Computation and 809 analysis: P.G., R.H., E.V.W., A.M.; Clinical input: N.M., A.L.D., E.J.; Resources and funding 810 acquisition: A.T.M., A.M., A.L.D., P.G., E.V.W.; Writing – Original draft: P.G., E.J., J.H., A.T.M.

## Competing interests

The authors declare no competing interests. None of the contents of this manuscript has been previously published or is under consideration elsewhere. All the authors have read and approved the final version of the manuscript before submission.
