## [Transparent Peer Review file · Nature Communications]

Evaluation of fetal exposure to environmental noise using a computer-generated model

Corresponding Author: Dr Pierre G lat

Version 0:

Reviewer comments:

Reviewer #1

(Remarks to the Author)

My expertise is in numerical analysis so I cannot comment on the significance of the work to the field of environmental noise impact. However, I can comment on the numerical simulation methodology. In my opinion this is sound. The BEM is a well-established tool for acoustic simulation that has been shown in many studies (both theoretical and empirical) to provide accurate and reliable results. The study takes advantage of many state-of-the-art techniques including matrix compression and OSRC preconditioning techniques. The authors have used an open-source BEM package and have provide codes and parameter choices so the study should be reproducible. The methodology has been validated on a simple test problem for which an exact solution is available, with convincing results. I recommend the paper for publication.

I have only two minor comments for improvement:

- Eqn (1) didn't get rendered properly, with what I suppose should be " $\tan(\delta)$ " being rendered as "ta ". This happened in both Adobe Reader and Sumatra.
- Fig 6 is supposed to show comparison with analytical solution, but I can only see 2 curves in each plot when I was expecting to see 4. Are the analytical solutions plotted in faint dashed lines? It's not clear, and neither the caption nor the legend explains things. This should be sorted out.

(Remarks on code availability)

No, but I am familiar with the BEMPP backend code and know it to be thoroughly tested and benchmarked.

Reviewer #2

(Remarks to the Author)

This important work uses an MRI based model of fetal/maternal anatomy to model acoustic noise levels reaching the fetus. The results show that sounds (especially lower frequencies) are attenuated "by as little as 6 dB."

An acknowledged limitation of the dataset was only examining fetuses over 30 weeks gestational age. This limits the generalizability of the results because the position, anatomy, tissue type fractions, etc of a younger fetus may impact results. If MRI of <30 weeks gestational age fetuses are not directly available to the authors, there may be freely available online repositories with MRI datasets of younger fetuses.

In the "results" section, the authors stated they "hypothesized" that the resolution (0.74 x 0.74 in plane) of the MRI was sufficient given the sound wavelength. Authors should consider changing the word "hypothesized" to "assumed" (or similar) given this hypothesis was not tested. I agree with the assumption that the MRI resolution is sufficient, however the full 3D resolution should be provided, not just the in-plane resolution. (Please see next comment on the MRI reconstruction).

The spatial resolution of the MRI scan and details of the MRI acquisition and reconstruction should be more completely reported. In the Methods section, it is stated that the T2-weighted HASTE was acquired in 3 orthogonal planes with slice thickness of 4mm. It is unclear if the final resolution of the used images are (0.74 x 0.74 x 4mm)? Or were all 3 orthogonal planes combined with a computer assisted reconstruction to generate a 0.74mm isotropic volume? If so, the procedure should be described and/or cited. Please clarify this in the methods section and in the results section.

One of the great advantages of integrating MRI into your model is the ability to visualize anatomy in combination with sound attenuation. However, the figures are lacking in clear anatomical landmarks or visual cues to orient the readers with the location of spine, abdomen, fetal head, etc. Fusion of unsegmented MR image slices in combination with at least some of the sound pressure level maps would aid in interpretation.

Is there any possible application of using MRI and your modeling method to provide individualized noise risk profiles?

(Remarks on code availability)

Version 1:

Reviewer comments:

Reviewer #1

(Remarks to the Author)

My comments have been addressed and I recommend publication.

(Remarks on code availability)

Reviewer #2

(Remarks to the Author)

The authors have addressed the reviewer comments.

(Remarks on code availability)

Dear Reviewers,

We greatly appreciate the time you have taken to read through and assess our manuscript and we are grateful for your constructive feedback and criticism. We are including responses to all your points, below. We are also uploading the revised manuscript with changes marked in red font. We hope that the manuscript is thereby improved. Furthermore, we have included higher resolution images in the revised version of the manuscript.

We have also made additional changes to comply with the Nature Communications formatting instructions, including those detailed below.

- We have renamed the datasets as Subjects 1, 2 etc. to improve readability.
- The word “novel” which appeared 3 times in the manuscript and which described the computational model as well as BEM formulation has been removed from the manuscript.
- Table 2 has been removed to comply with the upper limit of 10 display items. Bullet points have instead been opted for to list the MP3 filenames associated with the soundscapes.
- Figures 6 and 7 have been consolidated to enable us to include results on an additional dataset with a gestational age of less than 30 weeks.
- Other editorial improvements have been carried out.

Sincerely,

Pierre Gélât, Elwin Van 't Wout, Seyyed Reza Haqshenas, Andrew Melbourne, Anna L. David, Nada Mufti, Julian Henriques, Aude Thibaut De Maisieres and Eric Jauniaux

Response to Reviewer #1:

My expertise is in numerical analysis so I cannot comment on the significance of the work to the field of environmental noise impact. However, I can comment on the numerical simulation methodology. In my opinion this is sound. The BEM is a well-established tool for acoustic simulation that has been shown in many studies (both theoretical and empirical) to provide accurate and reliable results.

We would like to thank Reviewer 1 for this statement. We also believe this to be the case. Indeed, we have published results and validation on a range of biomedical ultrasound problems, as already highlighted in our manuscript.

The study takes advantage of many state-of-the-art techniques including matrix compression and OSRC preconditioning techniques. The authors have used an open-source BEM package and have provide codes and parameter choices so the study should be reproducible. The methodology has been validated on a simple test problem for which an exact solution is available, with convincing results. I recommend the paper for publication.

We have indeed used the legacy Bempp kernel which we have integrated into the OptimUS Python library, along with novel preconditioners and formulations which we have developed, which have made solving this problem possible. Our library, as well as Bempp, has been the subject of extensive validation published in peer reviewed journals as well as on our GitHub repository. We are pleased that the reviewer recommends our manuscript for publication.

I have only two minor comments for improvement:

- Eqn (1) didn't get rendered properly, with what I suppose should be " $\tan(\delta)$ " being rendered as "ta ". This happened in both Adobe Reader and Sumatra.

This indeed seems to have happened when converting to PDF. We have now remedied this.

- Fig 6 is supposed to show comparison with analytical solution, but I can only see 2 curves in each plot when I was expecting to see 4. Are the analytical solutions plotted in faint dashed lines? It's not clear, and neither the caption nor the legend explains things. This should be sorted out.

We agree that the dashed lines make it difficult to distinguish the analytical from the OptimUS solution. We have instead opted for dotted lines and we have also revised the figure legends so that both solutions are clearly identifiable. Note that we have consolidated figures 6 and 7 into figure 7 to allow for the additional dataset requested by Reviewer 2.

Reviewer #1 (Remarks on code availability):

No, but I am familiar with the BEMPP backend code and know it to be thoroughly tested and benchmarked.

Our code is indeed fully traceable and available on our GitHub repository, which is referenced in the manuscript: <https://github.com/optimuslib/optimus>

Response to Reviewer #2:

This important work uses an MRI based model of fetal/maternal anatomy to model acoustic noise levels reaching the fetus. The results show that sounds (especially lower frequencies) are attenuated "by as little as 6 dB."

We are very happy that Reviewer 2 acknowledges the importance of our work, which the authors put considerable time and effort into.

An acknowledged limitation of the dataset was only examining fetuses over 30 weeks gestational age. This limits the generalizability of the results because the position, anatomy, tissue type fractions, etc of a younger fetus may impact results. If MRI of <30 weeks gestational age fetuses are not directly available to the authors, there may be freely available online repositories with MRI datasets of younger fetuses.

We acknowledge that the manuscript can be improved by including a dataset with less than 30 weeks gestational age. We have sourced an MRI taken from an appropriately grown control pregnancy at 25 weeks and 1 day gestational age and generated meshes for the key anatomical groups considered in this study (abdomen, spine and uterus). We have repeated calculations of in utero sound propagation. Figure 2 now features frequency responses associated with this additional dataset and Figure 3 features sound maps at 5 kHz, 10 kHz and 20 kHz.

In the "results" section, the authors stated they "hypothesized" that the resolution (0.74 x 0.74 in plane) of the MRI was sufficient given the sound wavelength. Authors should consider changing the word "hypothesized" to "assumed" (or similar) given this hypothesis was not tested. I agree with the assumption that the MRI resolution is sufficient, however the full 3D resolution should be provided, not just the in-plane resolution. (Please see next comment on the MRI reconstruction).

We are happy that Reviewer 2 agrees that the MRI resolution is sufficient for acoustic calculations within the audio range. As suggested, we have changed "hypothesised" to "assumed" as this is indeed more relevant. We also acknowledge that we omitted the full resolution in the Results section under "Anatomical data", and we have now

provided this as 0.74 x 0.74 x 4 mm. It is indeed possible to combine imaging data by using super-resolution reconstruction methods to obtain higher through-plane resolution and we have published in this area. However, these algorithms are typically used on small regions of interest and due to the transforms applied, typically perform poorly across a region as large as the uterus, with artefacts away from the central field of view. For this work we have used the original data as we felt it was of sufficient resolution for the methods applied. Future work could indeed use SRR methods, and this might be one way, with validation, to increase precision in individualised noise-risk profiles.

The spatial resolution of the MRI scan and details of the MRI acquisition and reconstruction should be more completely reported. In the Methods section, it is stated that the T2-weighted HASTE was acquired in 3 orthogonal planes with slice thickness of 4mm. It is unclear if the final resolution of the used images are (0.74 x 0.74 x 4mm)? Or were all 3 orthogonal planes combined with a computer assisted reconstruction to generate a 0.74mm isotropic volume? If so, the procedure should be described and/or cited. Please clarify this in the methods section and in the results section.

We have now clarified the resolution for data acquisition as 0.74 x 0.74 x 4 mm and we did not use a super-resolution reconstruction in this work. We have also provided more detail of the MRI protocol in the Methods section under “MRI patient data segmentation”.

One of the great advantages of integrating MRI into your model is the ability to visualize anatomy in combination with sound attenuation. However, the figures are lacking in clear anatomical landmarks or visual cues to orient the readers with the location of spine, abdomen, fetal head, etc. Fusion of unsegmented MR image slices in combination with at least some of the sound pressure level maps would aid in interpretation.

We agree that there were insufficient anatomical cues in some of our figures to fully appreciate and interpret our results. We have now labelled the key anatomical groups (abdomen, spine and uterus) in Figure 1 (A-D) as well as in Figures 3-6 A. We have also labelled the associated intersections of these anatomical groups with the transverse plane passing through the uterus barycentre in Figures 3-6 D. We hope that the Figures are improved as a results of this.

Is there any possible application of using MRI and your modeling method to provide individualized noise risk profiles?

Whilst this is dependent on the sound source and position, as well as the patient specificity of tissue properties, this is an important point which we have now discussed.

We relate individual risk to noise sources whose spectral content overlaps with resonances and we discuss future research avenues.